# Minimum Curvature Manifold Learning

## Abstract

It is widely observed that vanilla autoencoders can have low manifold learning accuracy given a noisy or small training dataset. Recent work has discovered that it is important to regularize the decoder that explicitly parameterizes the manifold, where a neighborhood graph is employed for decoder regularization. However, one caveat of this method is that it is not always straightforward to construct a correct graph. Alternatively, one may consider naive graph-free regularization methods such as minimizing the norm of the decoder's Jacobian or Hessian, but these norms are not coordinate-invariant (i.e. reparametrization-invariant) and hence do not capture any meaningful geometric quantity of the manifold nor result in geometrically meaningful manifold regularization effects. Another recent work called the isometric regularization implicitly forces the manifold to have zero intrinsic curvature, resulting in some geometrically meaningful regularization effects. But, since the intrinsic curvature does not capture how the manifold is embedded in the data space from an extrinsic perspective, the regularization effects are often limited. In this paper, we propose a *minimum extrinsic curvature principle* for manifold regularization and **Minimum Curvature Autoencoder (MCAE)**, a graph-free coordinate-invariant extrinsic curvature minimization framework for autoencoder regularization. Experiments with various standard datasets demonstrate that MCAE improves manifold learning accuracy compared to existing methods, especially showing strong robustness to noise.

## 1 Introduction

Autoencoders are widely used to identify, given a set of high-dimensional data, the underlying lower-dimensional manifold structure and its coordinate space, simultaneously (Kramer, 1991). The decoder explicitly parameterizes the data manifold as a mapping from a lower-dimensional coordinate space (i.e., latent space) to the high-dimensional data space, and the encoder maps data points to their corresponding coordinates (i.e., latent values). However, vanilla autoencoders trained to reconstruct the given training data often learn manifolds that severely overfit to noisy training data or are wrong in regions where there are fewer data, impairing their manifold learning performances.

It has been recently discovered by Lee et al. (2021) that autoencoder regularization methods that focus on regularizing the latent space distributions determined entirely by the encoders (Kingma & Welling, 2013; Tolstikhin et al., 2018; Makhzani et al., 2015; Rifai et al., 2011) are not sufficient to learn correct manifolds, yet it is important to properly regularize the decoders that parameterize the manifolds. In (Lee et al., 2021), neighborhood graphs constructed from data are successfully utilized to regularize the local geometry and connectivity of the manifold, significantly improving the manifold learning accuracy. However, the underlying premise behind this method is that the graph has to be accurate, yet constructing a correct graph may not be always straightforward.

There are some graph-free methods such as the denoising autoencoder (Vincent et al., 2010) and reconstruction contractive autoencoder (Alain & Bengio, 2014) that regularize not only an encoder but also a decoder. They can learn manifolds that are robust to noise to some extent, but when the noise level is large, the performance is often less-than-desirable, and they do not always produce correct manifolds, especially in regions where there are fewer data (discussed in more detail in Section 4.2).

Since the decoder needs to be regularized, one may come up with some naive regularization strategies such as minimizing the norm of the decoder's Jacobian or Hessian, considering them as mea-

Same Manifold with Different Parameterizations | Zero Intrinsic Curvature Manifolds

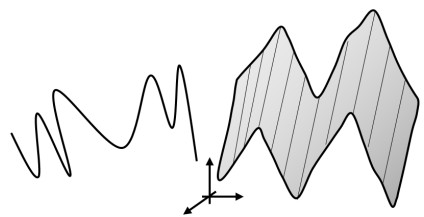

$$\|J_f\| > \|J_{f'}\|$$

Figure 1: *Left*: Two decoders $f$ and $f'$ parameterize the same data manifold where the norm of Jacobian of $f'$ is smaller than that of $f$, i.e., $\|J_f\| > \|J_{f'}\|$. *Right*: A curve and developable surface embedded in $\mathbb{R}^3$ have zero intrinsic curvatures.

sures of the manifold's smoothness. However, these norms do not properly capture any geometric quantity of the manifold because they are not reparametrization-invariant (or coordinate-invariant). As shown in Figure 1 (*Left*), just by increasing the volume of the latent space without actually changing the manifold, i.e., re-parametrizing the manifold $f \mapsto f'$, the above norms can be minimized.

Just recently, a coordinate-invariant geometric distortion measure has been introduced to regularize the decoder to be a geometry-preserving mapping, which is called the isometric regularization (LEE et al., 2022), so that the data space geometry is preserved in the latent space. Minimizing this distortion measure implicitly forces the learned manifold to have zero intrinsic curvature – which only depends on distances measured within the manifold (e.g., a cylinder's side surface has zero intrinsic curvature unlike the spherical surface) –, resulting in some geometrically meaningful manifold regularization effects.

The intrinsic curvature, however, does not capture how the manifold lies in the data space[1], and thus minimizing the manifold's intrinsic curvature may not be enough to learn correct manifolds. For example, curves and developable surfaces [2] in $\mathbb{R}^3$ always have zero intrinsic curvatures, e.g., Figure 1 (*Right*), regardless of how severely they are curved from an extrinsic point of view (Do Carmo, 2016).

The main contribution of this paper is a *coordinate-invariant extrinsic curvature minimization framework* for autoencoder regularization, which we refer to a **Minimum Curvature Autoencoder (MCAE)**, that is graph-free and effectively improves the manifold learning accuracy given a noisy or small training dataset. Specifically, we develop a coordinate-invariant extrinsic curvature measure of the learned manifold, by investigating how smoothly tangent space changes on the manifold, and use it as a regularization term.

To make things more explicit, let $\mathcal{M}$ be a manifold of dimension $m$ embedded in $\mathbb{R}^D$. Consider a mapping $T$ that maps a point x in $\mathcal{M}$ to its tangent space $T_x\mathcal{M}$, a linear subspace that has the dimension of $m$ attached at x, i.e., $T(x) = T_x\mathcal{M}$. The set of all linear subspaces of dimension $m$ in $\mathbb{R}^D$ forms a manifold called the Grassmann manifold denoted by $\mathrm{Gr}(m, \mathbb{R}^D)$ (Bendokat et al., 2020), and thus the mapping $T$ can be viewed as a mapping between two Riemannian manifolds, i.e., $T : \mathcal{M} \to \mathrm{Gr}(m, \mathbb{R}^D)$. By using the Dirichlet energy (Eells & Lemaire, 1978), a natural smoothness measure of mappings between two Riemannian manifolds defined in a coordinate-invariant way, we formulate an extrinsic curvature measure. We also propose a practical estimation strategy of the curvature measure that can be used for high-dimensional problems, reducing computation costs.

Experiments on diverse image and motion capture data confirm that, compared to existing graph-free regularized autoencoders, our MCAE improves manifold learning accuracy for noisy and small training datasets. In particular, our experiments show that even compared to the methods specially designed to be robust to input perturbations such as the DAE (Vincent et al., 2010) and RCAE (Alain & Bengio, 2014), the MCAE shows comparable or even in some cases significantly higher robust manifold learning performance.

---

[1]Manifold's intrinsic properties are defined without involving any embedding.

[2]A developable surface can be formed by bending or rolling a planar surface without stretching or tearing.

## 2 GEOMETRIC PRELIMINARIES

### 2.1 GRASSMANN MANIFOLD

In this section, we review the Grassmann manifold and its Riemannian geometry from a matrix-analytic perspective. The Grassmann manifold is defined as the set of all $m$ dimensional linear subspaces of the Euclidean space $\mathbb{R}^D$, denoted by $\mathrm{Gr}(m, \mathbb{R}^D)$; this can be identified with the set of orthogonal rank-$m$ projection matrices as follows:

$$\mathrm{Gr}(m, \mathbb{R}^D) = \{P \in \mathbb{R}^{D \times D} \mid P^T = P,\ P^2 = P,\ \mathrm{rank}(P) = m\}, \tag{1}$$

which is an $m(D - m)$ dimensional manifold; which associates $P \in \mathrm{Gr}(m, \mathbb{R}^D)$ with the linear subspace $\mathrm{range}(P) \subset \mathbb{R}^D$. This is an implicit parametrization of the Grassmann manifold considered as being embedded in the Euclidean space $\mathbb{R}^{D \times D}$. For more formal and detailed descriptions of the Grassmann manifold, we refer to (Bendokat et al., 2020).

Given a rank-$m$ matrix $J \in \mathbb{R}^{D \times m}$, one may want to consider its range, an $m$-dimensional linear subspace in $\mathbb{R}^D$, as an element of the Grassmann manifold. The embedding $E : \mathbb{R}^{D \times m} \to \mathrm{Gr}(m, \mathbb{R}^D)$ such that $E(J) = J(J^T J)^{-1} J^T$ properly converts $J$ to the element of (1). We note that (i) $\mathrm{range}(J) = \mathrm{range}(E(J))$ and (ii) $E(J) = E(JA)$ for any $m \times m$ invertible matrix $A \in \mathbb{R}^{m \times m}$ since the transformation $J \mapsto JA$ does not change the range.

Next, we introduce the basic Riemannian structure of the Grassmann manifold. At a point $P \in \mathrm{Gr}(m, \mathbb{R}^D)$, the tangent space is defined as follows:

$$T_P\mathrm{Gr}(m, \mathbb{R}^D) := \{V \in \mathbb{R}^{D \times D} \mid V^T = V,\ VP + PV = V\}, \tag{2}$$

which can be derived from (1) by differentiating the constraints. One canonical choice of the Riemannian metric is given as follows:

$$\langle V_1, V_2 \rangle := \frac{1}{\sqrt{2}} \mathrm{Tr}(V_1^T V_2) \quad \text{for} \quad V_1, V_2 \in T_P\mathrm{Gr}(m, \mathbb{R}^D). \tag{3}$$

This metric is invariant under the orthogonal transformation, i.e., $\langle V_1, V_2 \rangle = \langle RV_1, RV_2 \rangle$ for any $D \times D$ orthogonal matrix $R$.

### 2.2 DIRICHLET ENERGY FOR MAPPINGS BETWEEN RIEMANNIAN MANIFOLDS

This section introduces the Dirichlet energy for mappings between two Riemannian manifolds. Let $\mathcal{M}$ and $\mathcal{N}$ be Riemannian manifolds of dimension $m$ and $n$; we will consider a differentiable mapping f : $\mathcal{M} \to \mathcal{N}$. We will assume x $\in \mathcal{M}$ is explicitly parametrized by local coordinates as $x \in \mathbb{R}^m$ and the Riemannian metric at x $\in \mathcal{M}$ is expressed as $m \times m$ positive-definite matrix $G(x) = (g_{ij}(x)) \in \mathbb{R}^{m \times m}$, and $\mathcal{N}$ is embedded in the Euclidean space of higher dimension as $N \subset \mathbb{R}^d$ ($d \gg n$) and the Riemannian metric at y $\in \mathcal{N}$ is given as $\langle \cdot, \cdot \rangle_y$ for $y \in N$ (e.g. Grassmann manifold). The mapping f is expressed as $f : \mathbb{R}^m \to N \subset \mathbb{R}^d$ such that $y = f(x)$.

The Dirichlet energy, a global measure of how much the mapping $f$ changes, is defined as follows:

$$\int_{\mathcal{M}} \sum_{i=1}^{m} \sum_{j=1}^{m} g^{ij}(x) \langle \frac{\partial f}{\partial x^i}(x), \frac{\partial f}{\partial x^j}(x) \rangle_{f(x)} \sqrt{\det\ G(x)}\ dx^1 \cdots dx^m, \tag{4}$$

where $g^{ij}(x)$ denotes $(i, j)$-th element of the inverse of $G(x)$ and $\sqrt{\det\ G(x)}\ dx^1 \cdots dx^m$ is the Riemannian volume form, which corresponds to the integral functional from the theory of harmonic maps; this integral is an intrinsic quantity (i.e., coordinate-invariant). We note that the integrand is a local measure of how much the mapping $f$ changes. We refer to the extensive literature on the theory and applications of harmonic maps, e.g., (Eells & Lemaire, 1978; 1988; Park & Brockett, 1994; Jang et al., 2020; LEE et al., 2022).

## 3 MINIMUM CURVATURE AUTOENCODERS

In this section, we propose a regularized autoencoder based on the principle of minimum curvature manifold learning. Throughout, we consider a data space $\mathbb{R}^D$ and latent space $\mathbb{R}^m$ ($D \gg m$) and

denote a parametric encoder by $g_\phi : \mathbb{R}^D \to \mathbb{R}^m$ such that $z = g_\phi(x)$, and a parametric decoder by $f_\theta : \mathbb{R}^m \to \mathbb{R}^D$ such that $x = f_\theta(z)$. The manifold parametrized by the decoder will be denoted by $\mathcal{M}_\theta$, and the Jacobian of the decoder by $J_\theta(z) = \frac{\partial f_\theta}{\partial z}(z)$. Given a set of data points $\{x_i \in \mathbb{R}^D\}_{i=1}^N$, the empirical data distribution will be denoted by $\hat{p}(x) := \frac{1}{N} \sum_{i=1}^N \delta(x - x_i)$ and the latent space distribution encoded by $g_\phi$ by $\hat{p}_\phi(z) := \frac{1}{N} \sum_{i=1}^N \delta(z - g_\phi(x_i))$. The subscripts show what variables each function or geometric object depends on, either $\theta$ or $\phi$.

### 3.1 COORDINATE-INVARIANT EXTRINSIC CURVATURE MEASURE

In this section, we formulate a coordinate-invariant (i.e., reparametrization-invariant) extrinsic curvature measure of the manifold $\mathcal{M}_\theta$ embedded in $\mathbb{R}^D$. We begin by introducing the notion of coordinate-invariance:

**Definition 1.** *Given a manifold $\mathcal{M}$ of dimension $m$ embedded in $\mathbb{R}^D$, let $f : \mathbb{R}^m \to \mathcal{M}$ be its explicit parametrization. A functional $\mathcal{F}(f)$ is coordinate-invariant (i.e., reparametrization-invariant) if, given any invertible mapping or coordinate transformation (i.e., reparametrization) $h : \mathbb{R}^m \to \mathbb{R}^m$, $\mathcal{F}(f) = \mathcal{F}(f \circ h^{-1})$.*

The coordinate-invariance is necessary to properly measure any geometrically meaningful quantity of the manifold. For example, the integration of the Frobenius norm of $J_\theta$ in coordinate space $\mathbb{R}^m$ is not coordinate-invariant, and hence does not capture any geometrically meaningful quantity of $\mathcal{M}_\theta$.

Now, we define a coordinate-invariant extrinsic curvature measure of $\mathcal{M}_\theta$. The core idea is to define a local measure of the extrinsic curvature by measuring how fast the tangent space $T_x\mathcal{M}_\theta$ changes within the neighborhood of $x$, and then integrate it over the manifold to define a global curvature measure. For this purpose, let a pair of mappings, encoder $g_\phi$ and decoder $f_\theta$, be a coordinate system for $\mathcal{M}_\theta$, and consider a mapping $T : \mathbb{R}^m \to \mathrm{Gr}(m, \mathbb{R}^D)$ such that $T(z)$ is the element of the Grassmann manifold (1) whose range is equal to $T_{f_\theta(z)}\mathcal{M}_\theta$. We note that the range of the Jacobian matrix $J_\theta(z) \in \mathbb{R}^{D \times m}$ is $T_x\mathcal{M}_\theta$, hence, by using the embedding $E : \mathbb{R}^{D \times m} \to \mathrm{Gr}(m, \mathbb{R}^D)$ such that $E(J_\theta) := J_\theta(J_\theta^T J_\theta)^{-1} J_\theta^T$, we can explicitly write the mapping $T$ as $T(z) = E(J_\theta(z))$.

Let $\mathcal{M}_\theta$ be assigned with the Riemannian metric induced from the ambient space Euclidean metric, so that the metric expressed in the coordinate space is $J_\theta^T(z)J_\theta(z)$, and $\mathrm{Gr}(m, \mathbb{R}^D)$ be assigned with the Riemannian metric in (3). We use the dirichlet energy in (4) of the mapping $T$ as a coordinate-invariant extrinsic curvature measure, where the integral is replaced by the expectation over $\hat{p}_\phi(z)$:

**Definition 2.** *Given an encoder $g_\phi$, decoder $f_\theta$, and empirical distribution in coordinate space $\hat{p}_\phi(z)$, the global extrinsic curvature measure of $\mathcal{M}_\theta$ with respect to $\hat{p}_\phi(z)$ is defined as*

$$\mathcal{C}(\theta, \phi) := \mathbb{E}_{z \sim \hat{p}_\phi(z)} \Big[ \sum_{i=1}^m \sum_{j=1}^m (J_\theta^T J_\theta)_{ij}^{-1} Tr(\frac{\partial}{\partial z^i}(E(J_\theta)) \frac{\partial}{\partial z^j}(E(J_\theta))) \Big]. \tag{5}$$

**Proposition 1.** *The curvature measure $\mathcal{C}(\theta, \phi)$ in Definition 2 is coordinate-invariant, i.e., for another pair of encoder $g_{\phi'} := h \circ g_\phi$ and decoder $f_{\theta'} := f_\theta \circ h^{-1}$ with any invertible map or coordinate transformation $h$ such that $z' = h(z)$, the measure is invariant, i.e., $\mathcal{C}(\theta, \phi) = \mathcal{C}(\theta', \phi')$.*

*Proof.* The proof is given in the Appendix A.2 □

Our definition of the curvature generalizes classical definition of the curvature of a curve embedded in $\mathbb{R}^3$ from differential geometry (Kühnel, 2015) (please see Appendix A.3 for more details).

With the proposed curvature measure, we define a regularized autoencoder where the loss function consists of the following two terms i) reconstruction error term for manifold learning and ii) regularization term $\mathcal{C}(\theta, \phi)$ for curvature minimization:

$$\min_{\theta, \phi} \mathbb{E}_{x \sim \hat{p}(x)}[\|x - f_\theta \circ g_\phi(x)\|^2] + \alpha \, \mathcal{C}(\theta, \phi), \tag{6}$$

where $\alpha$ is the regularization coefficient, which we refer to as the **Minimum Curvature Autoencoder (MCAE)**.

## 3.2 PRACTICAL IMPLEMENTATIONS

This section introduces two practical strategies for computation of the curvature measure (5).

**Augmented Distribution:** In (5), the local curvature measure is expected over the empirical latent space distribution. However, the influence of the measure is then limited to regions where data is available; thus the manifold's curvature in regions where data is no data may not be properly regularized. In practice, we use data augmentation to resolve this issue. Following (Chen et al., 2020; LEE et al., 2022), we use the modified mix-up data-augmentation method with a parameter $\eta > 0$, where $\hat{p}_\phi(z)$ is augmented by $z = \delta z_1 + (1 - \delta)z_2$ such that $z_i \sim p_\phi(z), i = 1, 2$, where $\delta$ is uniformly sampled from $[-\eta, 1 + \eta]$. We set $\eta = 0.2$ throughout.

**Stochastic Trace Estimation:** At first glance, the curvature measure (5) seems computationally very expensive, because it involves the computation of the full Jacobian $J_\theta$ of a deep neural network and derivative of the Jacboaidn $\frac{\partial J_\theta}{\partial z}$, and we even need to backpropagate through them when using the standard stochastic gradient descent algorithms. To efficiently compute the measure in practice, we use the Hutchinson's trace estimator (Hutchinson, 1989), i.e., $\text{Tr}(A) = \mathbb{E}_{v \sim \mathcal{N}(0,I)}[v^T A v]$, then the curvature measure $\mathcal{C}(\theta, \phi)$ has the following expression:

$$\mathcal{C}(\theta, \phi) = \mathbb{E}_{z \sim \hat{p}_\phi(z), v \sim \mathcal{N}(0,I_m), w \sim \mathcal{N}(0,I_D)}[v^T \frac{\partial(w^T E(J_\theta))}{\partial z} \frac{\partial(E(J_\theta)w)}{\partial z} G_\theta^{-1} v], \qquad (7)$$

where $I_k$ is the $k \times k$ identity matrix and $G_\theta = J_\theta^T J_\theta$. To implement this computationally efficiently, we use the Jacobian-vector and vector-Jacboian products in multiple times: (i) for $E(J_\theta)w = J_\theta G_\theta^{-1} J_\theta^T w$, we first use the vector-Jacobian product for $J_\theta^T w$ and the Jacobian-vector product for $J_\theta(G_\theta^{-1} J_\theta^T w)$, and (ii) for $\frac{\partial(E(J_\theta)w)}{\partial z}v$ and $\frac{\partial(E(J_\theta)w)}{\partial z}(G_\theta^{-1}v)$, we use the Jacobian-vector products. These techniques make the computation of (5) tractable for high-dimensional complex problems. Surprisingly, for the estimation of (7), using one sample of $v$ and $w$ at each $z \sim \hat{p}_\phi(z)$ was sufficient to train MCAE in our later experiments. When the latent space is high-dimensional, the matrix inverse computation $G_\theta^{-1}$ takes up most of the computation time. Using an approximate inverse can significantly reduce the computation time, see the Appendix A.6.

## 4 EXPERIMENTS

### 4.1 PARAMETER SWEEP

We first provide an empirical study on the effect of the most important parameter of MCAE, the regularization coefficient $\alpha$. Intuitively, as $\alpha$ increases, the tendency to minimize the extrinsic curvature of the manifold becomes stronger, so the learned manifold will become closer to a linear subspace. And, if $\alpha$ is too small, the learned manifold will not be different from that of the vanilla autoencoder; hence it is important to select an appropriate value for $\alpha$ depending on the dataset.

Figure 2 shows how $\alpha$ affects the learned manifold in MCAE with two examples. In the upper figure, given noisy two-dimensional data points, we train MCAEs with one-dimensional latent spaces. In the lower figure, given sparse three-dimensional data points constrained on the 2-sphere $S^2 := \{x \in \mathbb{R}^3 \mid \|x\| = 1\}$, we train MCAEs with one-dimensional latent spaces, where the decoder outputs are normalized to be in $S^2$. As can be seen, $\alpha = 0.01$ and $\alpha = 0.0001$ are good values for the upper and lower examples, respectively. In practice, we can find the optimal value of $\alpha$ with a proper validation criteria (e.g., mean reconstruction error for validation data).

### 4.2 COMPARISON TO OTHER REGULARIZATION METHODS

In this section, we compare the proposed MCAE with other regularized autoencoders and highlight the differences. Please refer to Appendix A.1 for more detailed comparisons.

**Comparison to Isometrically Regularized Autoencoders:** In the Isometrically Regularized Autoencoder (IRAE) (LEE et al., 2022), the decoder is regularized to be a scaled isometry; similar to (6), a regularization term that measures how far $f_\theta$ from being a scaled isometry is added to the reconstruction error term with the regularization coefficient $\alpha$. This regularization implicitly forces the learned manifold $\mathcal{M}_\theta$ to have zero intrinsic curvature, but not the extrinsic curvature; therefore

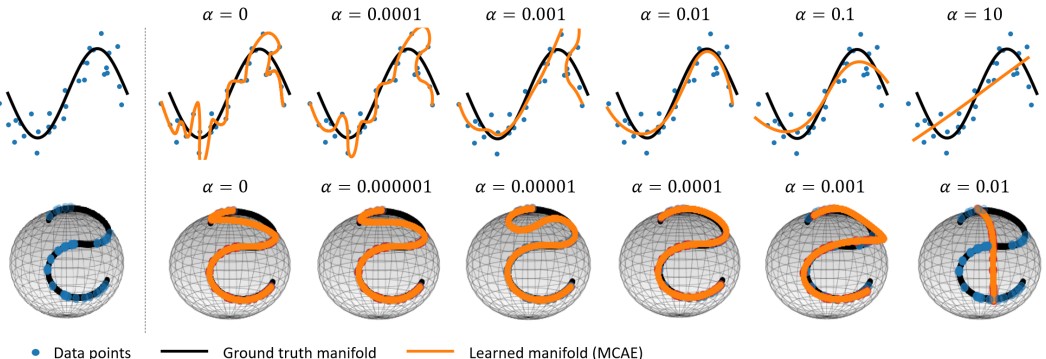

Figure 2: Learned manifold becomes flatter as the regularization coefficient $\alpha$ increases. *Upper*: Learned data manifolds of 1d sin-curve and noisy training data points. *Lower*: Learned data manifolds of 1d S-curve projected to the 2-sphere and sparse training data points.

it is at first glance expected that, when learning a one-dimensional manifold, the IRVE should not have any meaningful manifold regularization effect (since one-dimensional manifolds always have zero intrinsic curvatures).

Counterintuitively, as shown in Figure 3(a), our experiments show that the extrinsic curvature of the one-dimensional manifold learned by IRAE decreases as $\alpha$ increases. If the decoder's hypothesis space was a set of arbitrary smooth functions, this result would not have been obtained, but since the hypothesis space defined as the set of neural networks is smaller, the isometric regularization seems to reduce the extrinsic curvature at the expense of obtaining the isometric representations. Figure 3(b) shows how the reconstruction MSE for clean test data varies as a function of the extrinsic curvature of the learned manifold by IRAE and MCAE. As the curvature decreases or the regularization coefficient increases (from left to right), the test reconstruction MSE decreases, reaches a minimum, and then increases again. We note that the graph of MCAE lies lower than that of IRAE, implying that the MCAE can learn a more accurate manifold than the IRAE.

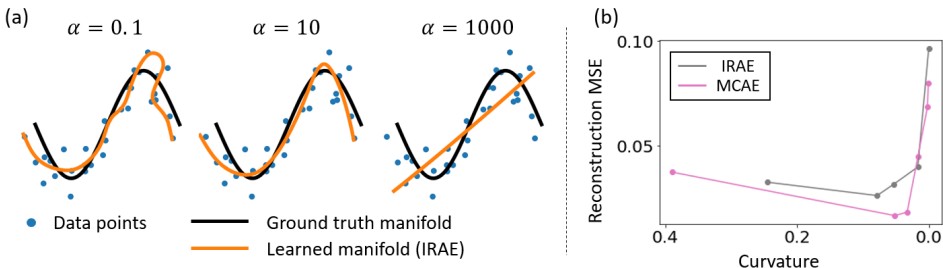

Figure 3: (a) Learned manifold by IRAE becomes flatter as the regularization coefficient $\alpha$ increases. (b) Test data reconstruction MSE (i.e., manifold learning accuracy) as a function of the extrinsic curvature obtained by IRAE and MCAE.

**Comparison to Denoising and Reconstruction Contractive Autoencoders:** Denoising Autoencoder (DAE) (Vincent et al., 2010) and Reconstruction Contractive Autoencoder (RCAE) (Alain & Bengio, 2014) are intuitive and straightforward regularization methods for learning manifolds robust to input perturbations. As shown in Figure 4 (*Upper*), the DAE and RCAE learn manifolds robust to noise to some extent. However, as shown in Figure 4 (*Lower*), for the projected S-curve example in Figure 2 (*Lower*), they still learn wrong manifolds in regions where there are fewer data and do not improve the vanilla autoencoder. On the other hand, the MCAE explicitly regularizes the learned manifold to have a small curvature globally and improves the manifold learning accuracy.

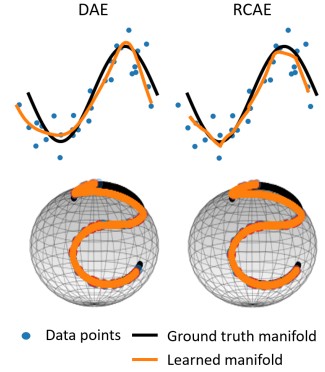

Figure 4: Learning by DAE and RCAE for examples in Figure 2.

Table 1: Averages and standard errors of the test data set reconstruction MSEs (5 times run) for the sincurve example in Figure 2 (*Upper*) with various Gaussian noise of standard deviations $0.1, 0.2, 0.3$, the lower the better. The best results are marked in bold. The numbers are written in units of $10^{-3}$.

| Noise | AE | VAE | DAE | RCAE | DCAE | DHAE | IRAE | MCAE |
|---|---|---|---|---|---|---|---|---|
| 0.1 | $3.98 \pm 0.22$ | $2.05 \pm 0.36$ | $2.23 \pm 0.26$ | $2.95 \pm 0.44$ | $2.81 \pm 0.21$ | $2.68 \pm 0.31$ | $1.63 \pm 0.25$ | $\mathbf{1.28 \pm 0.13}$ |
| 0.2 | $22.7 \pm 2.9$ | $6.34 \pm 0.72$ | $6.99 \pm 1.04$ | $12.5 \pm 1.1$ | $10.9 \pm 1.02$ | $14.9 \pm 2.87$ | $6.59 \pm 1.09$ | $\mathbf{4.56 \pm 0.69}$ |
| 0.3 | $68.5 \pm 17.3$ | $13.5 \pm 2.3$ | $17.8 \pm 3.4$ | $30.5 \pm 7.9$ | $30.7 \pm 4.8$ | $46.9 \pm 12.5$ | $20.2 \pm 5.1$ | $\mathbf{9.80 \pm 1.55}$ |

**Quantitative Comparisons of Noise Robustness:** As seen from the above examples, besides the proposed MCAE, the IRAE, DAE, RCAE all have the robustness properties to noise. We quantitatively compare the robust manifold learning performance given noisy input training data with the sincurve example in Figure 2 (*Upper*) with various noise levels, i.e., Gaussian noise with standard deviations of $0.1, 0.2, 0.3$. In addition to the IRAE, DAE, RCAE, we compare the MCAE with the vanilla Autoencoder (AE) and other regularized autoencoders such as the Variational Autoencoder (VAE) (Kingma & Welling, 2013), Decoder Contractive Autoencoder (DCAE), and Decoder Hessian Contractive Autoencoder (DHAE), where the DCAE and DHAE minimize the decoder's Jacobian norm and the decoder's Hessian norm, respectively. Table 1 shows the averages and standard errors of the test data set reconstruction MSEs, the lower the better. The MCAE produces the lowest errors regardless of the noise level.

### 4.3 IMAGE DATA

**Grayscale Image:** First, we investigate the manifold learning performance of MCAE compared to the other regularized autoencoders with the standard grayscale image data (*MNIST, Fashion-MNIST, KMNIST*) as the number of training + validation data and noise level varies. We use two-layer fully connected neural networks (512 nodes per layer) for both encoder and decoder with ELU activation functions, and the latent space dimensions are $16, 32, 32$, respectively.

Figure 6 shows the test reconstruction MSEs as a function of the number of training (80%) + validation (20%) data. For all methods, the error decreases as the number of data increases; MCAEs mostly produce the lowest errors except for some MNIST cases. Figure 7 shows the Peak Signal-to-Noise Ratios (PSNRs) computed with the clean test set data (the higher the better) as a function of the standard deviation of the Gaussian noise added to the

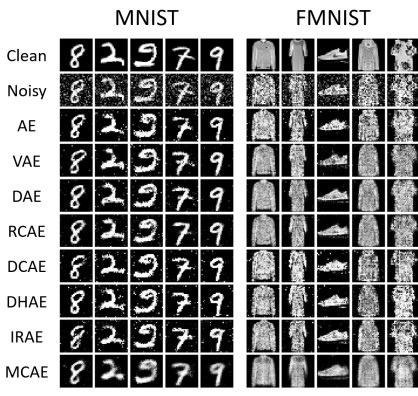

Figure 5: De-noising examples (noise level 0.3).

training data (the number of training data is 8000). The PSNR decreases as the noise level increases; MCAEs mostly produce the highest PSNRs. Figure 5 shows some de-noising examples with corrupted input data of MNIST and FMNIST.

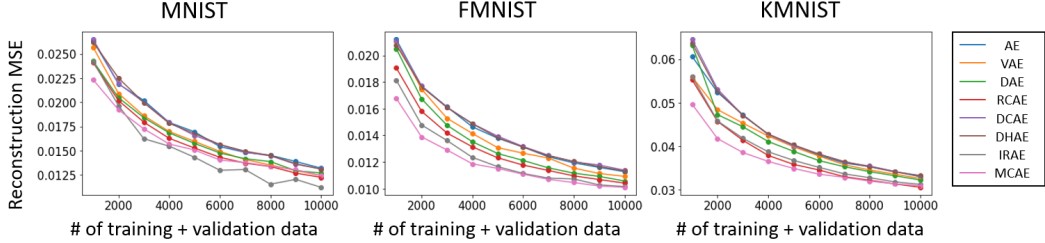

Figure 6: Test set MSEs as a function of the number of training (80%) + validation (20%) data, the lower the better.

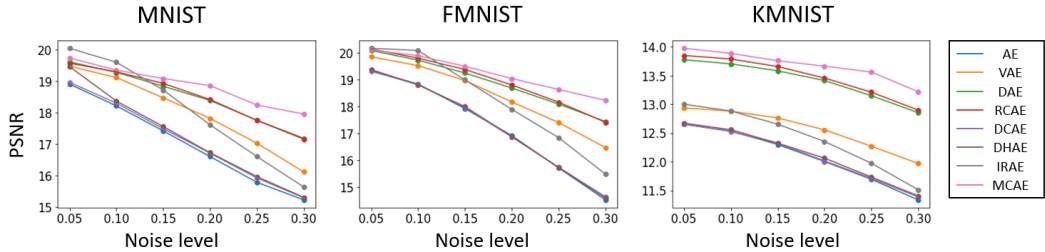

Figure 7: Test set Peak Signal-to-Noise Ratios (PSNR) as a function of the noise level, the higher the better.

**SVHN & CIFAR10 Image:** We compare the manifold learning performances of MCAE with other regularized autoencoders for the SVHN and CIFAR10 image datasets for both clean and corrupted training datasets. We use the convolutional and transposed convolutional neural networks for encoder and decoder with ReLU activation functions and the latent space dimensions are 64; the number of training data is 8000. For the corrupted training dataset cases, we add three different types of noise: (i) Gaussian, (ii) Shot, and (iii) Impulse noises adopted from (Hendrycks & Dietterich, 2019); see Figure 8.

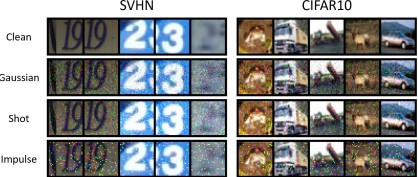

Figure 8: Corrupted SVHN and CI-FAR10 images.

Table 2 shows the test set MSEs for experiments with the clean training datasets, and Table 3 shows the PSNRs for experiments with the corrupted training datasets, where in both cases the metrics are computed with the clean test data. From the results, we note that (i) MCAE shows the second or third best results, (ii) MCAE does not improve the vanilla AE for the SVHN clean training dataset case, and (iii) for the corrupted training dataset cases, RCAE produces better results than the MCAE unlike the grayscale image data. Overall, compared to the grayscale image data, the minimum curvature regularization is less effective for SVHN and CIFAR10. One possible interpretation is related to the limitation of MCAE (discussed in the conclusion section), that the SVHN and CIFAR10 manifolds have locally very different curvatures and thus it is difficult to find a proper constant regularization co-

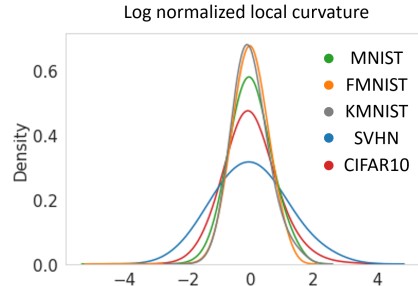

Figure 9: Density plots of the log-normalized local curvatures of manifolds learned by vanilla AEs.

efficient $\alpha$ in (6), because if we use a big enough $\alpha$ to correctly learn low curvature areas of the manifold, then high curvature areas can be overly flattened, and vice versa. Figure 9 shows the density plots of the log normalized local curvature of the learned manifolds by vanilla autoencoders, i.e., $\log(\kappa_i) - \overline{\log(\kappa)}, i = 1, \ldots, N$ where $\kappa_i$ is the local curvature at $i$-th training data points and $\overline{\log(\kappa)} = 1/N \sum_i \log(\kappa_i)$, which is invariant to the scale of the mean curvature. As shown in Figure 9, the variance of the SVHN manifold's local curvature is bigger than those of the others, which supports the above interpretation.

Table 2: Test set MSEs of autoencoders trained with clean datasets, the lower, the better. The best and second best results are marked in red and blue, respectively.

| Dataset | AE | VAE | DAE | RCAE | DCAE | DHAE | IRAE | MCAE |
|---------|-----|-----|-----|------|------|------|------|------|
| SVHN | 0.00228 | 0.00461 | 0.00228 | 0.00252 | 0.00255 | 0.00233 | 0.00213 | 0.00229 |
| CIFAR10 | 0.01204 | 0.01533 | 0.01204 | 0.01119 | 0.01303 | 0.01244 | 0.01176 | 0.01125 |

## 4.4 HUMAN SKELETON POSE DATA

In this section, we evaluate the MCAE with the human skeleton pose data adopted from the NTU RGB+D dataset (Shahroudy et al., 2016). A human pose skeleton data onsists of 25 three-dimensional key points and thus is considered a 75-dimensional vector. There are 60 different action

Table 3: Test data set PSNRs with various noise types, the higher, the better. The best and second best results are marked in red and blue, respectively.

| Dataset | Noise type | AE | VAE | DAE | RCAE | DCAE | DHAE | IRAE | MCAE |
|---------|-----------|-----|-----|-----|------|------|------|------|------|
| SVHN | Gaussian | 20.13 | 22.60 | 22.04 | 25.39 | 20.92 | 19.87 | 20.74 | 24.37 |
|  | Shot | 21.11 | 22.49 | 22.97 | 26.20 | 22.41 | 21.23 | 25.18 | 24.73 |
|  | Impulse | 19.33 | 19.06 | 20.28 | 23.31 | 19.18 | 19.25 | 19.71 | 20.18 |
| CIFAR10 | Gaussian | 17.06 | 17.60 | 17.78 | 19.51 | 17.10 | 16.92 | 17.35 | 18.62 |
|  | Shot | 17.18 | 17.46 | 17.87 | 19.52 | 17.26 | 17.18 | 18.49 | 18.64 |
|  | Impulse | 16.71 | 16.03 | 16.93 | 18.49 | 16.58 | 16.59 | 16.62 | 17.04 |

classes (e.g., drinking water, brushing teeth), and each action data consists of a sequence of skeleton poses. For each action class, we use randomly-selected 800 and 200 skeleton poses as training and validation data, and 9000 poses as test data. We use two-layer fully connected neural networks (512 nodes per layer) for both encoder and decoder with ELU activation functions, and the latent space dimension is 8.

Table 4 shows the averages and standard errors of the test data set reconstruction MSEs over 60 different action classes, the lower the better. MCAE mostly produces the lowest errors, especially by a significant margin for noisy training data cases. Figure 10 shows some example reconstruction results of noisy input skeleton data (noise level 0.05); MCAE shows the best de-noising results.

Table 4: Averages and standard errors of the test data set reconstruction MSEs with various Gaussian noise of standard deviations 0.05, 0.1, the lower the better. The best and second best results are marked in red and blue, respectively. The numbers are written in units of $10^{-3}$.

| Noise | AE | VAE | DAE | RCAE | DCAE | DHAE | IRAE | MCAE |
|-------|-----|-----|-----|------|------|------|------|------|
| 0 | $2.23 \pm 0.09$ | $2.95 \pm 0.13$ | $2.21 \pm 0.09$ | $2.17 \pm 0.09$ | $2.25 \pm 0.09$ | $2.22 \pm 0.09$ | $2.08 \pm 0.09$ | $2.11 \pm 0.09$ |
| 0.05 | $4.60 \pm 0.04$ | $4.32 \pm 0.02$ | $2.70 \pm 0.01$ | $2.98 \pm 0.01$ | $3.92 \pm 0.03$ | $4.07 \pm 0.03$ | $2.93 \pm 0.02$ | $2.20 \pm 0.01$ |
| 0.1 | $15.3 \pm 0.2$ | $13.5 \pm 0.2$ | $5.54 \pm 0.17$ | $7.50 \pm 0.16$ | $11.7 \pm 0.2$ | $12.5 \pm 0.2$ | $12.7 \pm 0.2$ | $3.09 \pm 0.15$ |

Figure 10: Human skeleton pose de-noising examples obtained by reconstructing noisy input data (noise level 0.05). Example poses are from the action class "eat meal".

## 5 CONCLUSION

In this paper, we have proposed a *minimum extrinsic curvature principle* for manifold regularization and developed a **Minimum Curvature Autoencoder (MCAE)**, by formulating a coordinate-invariant (reparametrization-invariant) hence geometrically correct extrinsic curvature measure. Our experiments show that the minimum curvature regularization can improve manifold learning accuracy for both noisy and small training datasets. The degree to which the performance is improved depends on the datasets, and especially for the grayscale image and human skeleton pose datasets, the MCAE outperforms the existing methods by a significant margin.

**Limitations and Future Directions**: In the current implementation of MCAE, the manifold's extrinsic curvature is minimized globally by using equal weights for all points. However, for manifolds that have locally very different curvatures, it is difficult to find a proper weight parameter $\alpha$ in (6). Ideally, low and high curvature areas of the manifold need to be regularized with higher and lower weights, respectively. By exploiting local curvature estimation algorithms, e.g., diffusion-based method (Bhaskar et al., 2022), developing a curvature regularization method with different local weights will be an interesting future research direction.

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

## A  APPENDIX

The appendix is organized as follows: (A.1) Related Works, (A.2) Proof of Proposition 1, (A.3) On the Extrinsic Curvature, (A.4) Experiment Details, (A.5) Additional Experiment Results, and (A.6) Computational Complexity.

### A.1  RELATED WORKS: REGULARIZED AUTOENCODERS

The framework of autoencoding together with the recent advances in deep learning techniques used for approximating arbitrary complex functions successfully addresses the manifold learning problem (Kramer, 1991). The core idea is to learn two mappings an encoder $g : \mathbb{R}^D \to \mathbb{R}^m$ and a decoder $f : \mathbb{R}^m \to \mathbb{R}^D$ approximated with deep neural networks so that the composition of them reconstructs the given data points $x_i \in \mathbb{R}^D$, i.e., $f \circ g(x_i) \approx x_i$, for $i = 1, \cdots, N$, and that the data points approximately lie on the image of the decoder, which we refer to as the learned manifold.

Many existing autoencoder regularization methods have focused on the representation learning perspective of autoencoders and studied how to regularize the latent space distributions for purposes like sampling, topology and geometry preserving, clustering, or capturing hierarchical structure (Rifai et al., 2011; Kingma & Welling, 2013; Wang et al., 2014; Makhzani et al., 2015; Tolstikhin et al., 2017; Chen et al., 2016; Tomczak & Welling, 2018; Klushyn et al., 2019; Moor et al., 2020; Schönenberger et al.; Duque et al., 2020; Chen et al., 2021); since the latent space distributions are entirely determined by the encoders, they mostly focus on regularizing the encoders but not decoders.

As discovered in (Lee et al., 2021), to learn the accurate manifold in the presence of data noise or given a small number of training data, regularization of the decoder is indeed more important,

because it is the decoder that has information about how the manifold lies in the data space. Based on the intuition that a local approximation of the decoder contains local geometric information on the decoded manifold (i.e. learned manifold), e.g., a local linear approximation of $f$ spans the tangent space, a priori constructed neighborhood graph is employed to regularize the local approximation of the decoder and hence the decoded manifold. This has shown improved manifold learning accuracy for both noisy and small training dataset cases, however obviously, the performance largely depends on the quality of the graph as in many other graph-based methods.

There are graph-free autoencoder regularization methods that regularize not only an encoder but also the decoder. Denoising autoencoder (Vincent et al., 2010) is trained to reconstruct a corrupted input to its clean version with the following loss

$$\sum_{i=1}^{N} \|x_i - f(g(x_i + \epsilon))\|^2, \tag{8}$$

for some noise variable $\epsilon$. As a limit case in (Alain & Bengio, 2014), the Jacobian of the reconstruction function is minimized where the loss is defined as follows:

$$\sum_{i=1}^{N} \|x_i - f(g(x_i))\|^2 + \alpha \left\|\frac{\partial f \circ g}{\partial x}(x_i)\right\|_F^2, \tag{9}$$

where $\alpha$ is the regularization coefficient and $\|\cdot\|_F$ denotes the Frobenius norm. These by construction attempt to learn manifolds robust to noise, but we note that (i) they are designed to be robust to noise during inference after being trained with clean data, but if training data points themselves are noisy, the robust manifold learning performance decreases and (ii) their regularization effects are limited to where data points are available.

Since regularizing the decoder that explicitly parameterizes the manifold is important, one may consider minimizing the norm of decoder's Jacobian as

$$\sum_{i=1}^{N} \|x_i - f(g(x_i))\|^2 + \alpha \left\|\frac{\partial f}{\partial z}(g(x_i))\right\|_F^2 \tag{10}$$

with the regularization coefficient $\alpha$ or the norm of decoder's Hessian $\sum_{i,j} \left\|\frac{\partial^2 f}{\partial z_i \partial z_j}(g(x_i))\right\|^2$. However, these norms do not capture geometric quantities of the learned manifold because they are not coordinate-invariant or reparmetrization-invariant, and thus they do not produce any meaningful regularization effects.

For example, consider a coordinate transformation $z' = h(z)$ which converts the encoder as $g \mapsto g' = h \circ g$ and decoder as $f \mapsto f' = f \circ h^{-1}$. The reconstruction loss is invariant since $f \circ g = f' \circ g'$, and hence the learned manifold is invariant, but the regularization term, the norm of decoder's Jacobian, is different:

$$\left\|\frac{\partial f'}{\partial z'}(g'(x_i))\right\|_F^2 = \left\|\frac{\partial f}{\partial z}(g(x_i))\frac{\partial h^{-1}}{\partial z'}(g'(x_i))\right\|_F^2 \neq \left\|\frac{\partial f}{\partial z}(g(x_i))\right\|_F^2. \tag{11}$$

This implies that we can minimize the norm of decoder's Jacobian just by increasing the norm of Jacobian of $h^{-1}$ without actually changing the learned manifold. A similar argument holds for the Hessian norm.

Recent works (Chen et al., 2020; LEE et al., 2022) have suggested decoder regularization methods for learning isometric representations that preserve geometry of the data space. A common goal is to learn a decoder $f : \mathbb{R}^m \to \mathbb{R}^D$ that satisfies

$$\frac{\partial f}{\partial z}(z)^T \frac{\partial f}{\partial z}(z) = cI \text{ for all } z \in \nu(\mathbb{R}^m) \tag{12}$$

for some positive scalar $c$, where $I$ is the $m \times m$ identity matrix and $\nu(\mathbb{R}^m)$ is the support of the latent space data distribution. Such mappings are formally defined as scaled isometries in (LEE et al., 2022), which are geometry-preserving mappings in the sense that latent space straight lines are mapped to the geodesic curves in the learned manifold.

Regularizing the decoder to be a scaled isometry, beyond finding geometry-preserving representations, has an implicit manifold regularization effect. According to Gauss's Theorema Egregium

which states that "The Gaussian curvature of a surface is invariant under local isometry", for scaled isometries $f$ to exist, the Gaussian curvature of the learned manifold should be the same as that of the Euclidean space (i.e., zero). In other words, it has an *implicit intrinsic curvature minimization* effect, which is different from the method proposed in this paper that explicitly minimizes the extrinsic curvature.

## A.2 PROOF OF PROPOSITION 1

*Proof.* Let's denote by

$$c(\theta, \phi) = \sum_{i,j} (J_\theta^T J_\theta)_{ij}^{-1} \text{Tr}(\frac{\partial E(J_\theta)}{\partial z_i} \frac{\partial E(J_\theta)}{\partial z_j}).$$

Given a coordinate transformation $z' = h(z)$ that maps $(g_\phi, f_\theta) \mapsto (g_{\phi'}, f_{\theta'}) = (h \circ g_\phi, f_\theta \circ h^{-1})$, the following transformation rules hold: $J_\theta \mapsto J_{\theta'} = J_\theta \cdot \frac{\partial h^{-1}}{\partial z'}$ and $\frac{\partial I}{\partial z} \mapsto \frac{\partial I}{\partial z'} = \frac{\partial I}{\partial z} \frac{\partial h^{-1}}{\partial z'}$ for some scalar-valued function $I(z)$. We note that, since $E(J) = E(JA)$ for some arbitrary invertible matrix $A$, the embedding is invariant, i.e., $E(J_{\theta'}) = E(J_\theta)$. Let $I_{\alpha\beta}$ denote the $(\alpha, \beta)$-component of $E(J_\theta)$, then, by using $\text{Tr}(AB) = \sum_{\alpha,\beta} A_{\alpha\beta} B_{\beta\alpha}$ and denoting $\frac{\partial h^{-1}}{\partial z'}$ by $H$,

$$
\begin{aligned}
c(\theta, \phi) \mapsto c(\theta', \phi') &= \sum_{i,j} (J_{\theta'}^T J_{\theta'})_{ij}^{-1} \sum_{\alpha,\beta} \frac{\partial I_{\alpha\beta}}{\partial z_i'} \frac{\partial I_{\beta\alpha}}{\partial z_j'} \\
&= \sum_{\alpha,\beta} \frac{\partial I_{\alpha\beta}}{\partial z'} (J_{\theta'}^T J_{\theta'})^{-1} \left(\frac{\partial I_{\beta\alpha}}{\partial z'}\right)^T \\
&= \sum_{\alpha,\beta} \frac{\partial I_{\alpha\beta}}{\partial z} H (H^T J_\theta^T J_\theta H)^{-1} H^T \left(\frac{\partial I_{\beta\alpha}}{\partial z}\right)^T \\
&= \sum_{\alpha,\beta} \frac{\partial I_{\alpha\beta}}{\partial z} (J_\theta^T J_\theta)^{-1} \left(\frac{\partial I_{\beta\alpha}}{\partial z}\right)^T \\
&= \sum_{i,j} (J_\theta^T J_\theta)_{ij}^{-1} \sum_{\alpha,\beta} \frac{\partial I_{\alpha\beta}}{\partial z_i} \frac{\partial I_{\beta\alpha}}{\partial z_j} = c(\theta, \phi).
\end{aligned}
\tag{13}
$$

$\square$

## A.3 ON THE EXTRINSIC CURVATURE MEASURE

In this section, we we will derive the expression of our extrinsic curvature measure

$$\sum_{i,j} (J^T J)_{ij}^{-1} \text{Tr}(\frac{\partial J(J^T J)J^T}{\partial z^i} \frac{\partial J(J^T J)J^T}{\partial z^j})$$

for a one-dimensional manifold, i.e., a curve, embedded in $\mathbb{R}^D$. Let $x : \mathbb{R} \to \mathbb{R}^D$ be a smooth curve and assume that it is parameterized by arc-length, i.e., $\|\frac{\partial x}{\partial z}\| = J^T J = 1$. Then, the curvature becomes

$$
\begin{aligned}
\text{Tr}((\frac{\partial J J^T}{\partial z})^2) &= \text{Tr}((\frac{\partial}{\partial z}(\frac{\partial x}{\partial z} \frac{\partial x}{\partial z}^T))^2) \\
&= \text{Tr}((\frac{\partial^2 x}{\partial z^2} \frac{\partial x}{\partial z}^T + \frac{\partial x}{\partial z} \frac{\partial^2 x}{\partial z^2}^T)^2) \\
&= \text{Tr}(\frac{\partial^2 x}{\partial z^2} \frac{\partial x}{\partial z}^T \frac{\partial^2 x}{\partial z^2} \frac{\partial x}{\partial z}^T + 2 \frac{\partial^2 x}{\partial z^2} \frac{\partial x}{\partial z}^T \frac{\partial x}{\partial z} \frac{\partial^2 x}{\partial z^2}^T + \frac{\partial x}{\partial z} \frac{\partial^2 x}{\partial z^2}^T \frac{\partial x}{\partial z} \frac{\partial^2 x}{\partial z^2}^T) \\
&= \frac{\partial x}{\partial z}^T \frac{\partial^2 x}{\partial z^2} \frac{\partial x}{\partial z}^T \frac{\partial^2 x}{\partial z^2} + 2 \frac{\partial x}{\partial z}^T \frac{\partial x}{\partial z} \frac{\partial^2 x}{\partial z^2}^T \frac{\partial^2 x}{\partial z^2} + \frac{\partial^2 x}{\partial z^2}^T \frac{\partial x}{\partial z} \frac{\partial^2 x}{\partial z^2}^T \frac{\partial x}{\partial z} \\
&= 2(\frac{\partial x}{\partial z}^T \frac{\partial^2 x}{\partial z^2})^2 + 2 \frac{\partial^2 x}{\partial z^2}^T \frac{\partial^2 x}{\partial z^2}.
\end{aligned}
\tag{14}
$$

Since $\frac{\partial}{\partial z}\|\frac{\partial x}{\partial z}\| = 0$ implies that $\frac{\partial x}{\partial z}^T \frac{\partial^2 x}{\partial z^2} = 0$, our curvature measure for an arc-length parameterized curve $x(z)$ is simplified to $2\|\frac{\partial^2 x}{\partial z^2}\|$ that is twice the norm of second derivative. This is equivalent to the classical definition of the curvature of a curve.

## A.4 EXPERIMENT DETAILS

**Grayscale Image Data:** The image size is $28 \times 28$ and the pixel values are normalized between 0 and 1. The encoder and decoder are two-layer fully connected neural networks with the ELU activation functions and 512 nodes for each layer. The output layer is linear for the encoder and sigmoid for the decoder. For clean dataset cases, we use the following early stopping criteria in training: we stop the training if the mean reconstruction error for the validation dataset increases 10 times in a row; then we use the best model (i.e. the lowest validation errors) for evaluation. For noisy dataset cases, assuming that we don't have an access to the clean dataset during training, we do not use the early stopping and trained the model for a sufficiently big number of epochs for convergence (the number of epochs is 1000). The number of test data is 60000. For evaluation, we use clean test data for noisy training dataset cases as well. The batch size is 100 and the learning rate is 0.001.

**SVHN & CIFAR10 Image Data:** The image size is $32 \times 32$ and the pixel values are normalized between 0 and 1. For noisy training dataset experiments, we add noises as follows: (i) for Gaussian noise, the standard deviation is 0.1, (ii) for Shot noise, we multiply 0.15 to noise variables sampled from the Poisson distributions where $\lambda$ are image pixel values, and (iii) for Impulse noise, with $5\%$ probability we randomly add 1 to each pixel. The encoder and decoder are convolutional and transposed convolutional neural networks with the ReLU activation functions, where, denoting a convolution layer of input channel size $c_i$, output channel size $c_o$, kernel size $k$, stride $s$, and padding $p$ by Conv2d($c_i, c_o, k, s, p$) and transposed convolution layer by ConvTrans2d($c_i, c_o, k, s, p$), the following sequence of layers Conv2d(3, 128, 4, 2)-Conv2d(128, 256, 4, 2)-Conv2d(256, 512, 4, 2)-Conv2d(512, 1024, 2, 2)-Conv2d(1024, 64, 1) is used for encoder and ConvTrans2d(64, 1024, 8)-ConvTrans2d(1024, 512, 4, 2, 1)-ConvTrans2d(512, 256, 4, 2, 1)-ConvTrans2d(512, 3, 1) for decoder. The output layer is linear for the encoder and sigmoid for the decoder. For clean dataset cases, we use the following early stopping criteria in training: we stop the training if the mean reconstruction error for the validation dataset increases 10 times in a row; then we use the best model (i.e. the lowest validation errors) for evaluation. For noisy dataset cases, assuming that we don't have an access to the clean dataset during training, we do not use the early stopping and trained the model for a sufficiently big number of epochs for convergence (the number of epochs is 100). The number of test data is 63257 for SVHN and 10000 for CIFAR10. For evaluation, we use clean test data for noisy training dataset cases as well. The batch size is 8 and the learning rate is 0.0001.

**Human Skeleton Pose Data:** From the NTU RGB+D dataset, a set of human pose skeleton data that consists of 25 key points is extracted and pre-processed to be aligned. Specifically, 10000 poses are extracted from each action class (a total of 60 action classes is used), and they are rotated and translated so that the 1-2 key points direction becomes z-axis and 1-13 key points direction becomes the y-axis and the key point number 2 becomes the origin. The encoder and decoder are two-layer fully connected neural networks with the ELU activation functions and 512 nodes for each layer. The output layers are linear for both the encoder and decoder. For clean dataset cases, we use the following early stopping criteria in training: we stop the training if the mean reconstruction error for the validation dataset increases 10 times in a row; then we use the best model (i.e. the lowest validation errors) for evaluation. For noisy dataset cases, assuming that we don't have an access to the clean dataset during training, we do not use the early stopping and trained the model for a sufficiently big number of epochs for convergence (the number of epochs is 5000). The number of test data is 9000. For evaluation, we use clean test data for noisy training dataset cases as well. The batch size is 100 and the learning rate is 0.0001.

## A.5 ADDITIONAL EXPERIMENT RESULTS

**More Qualitative Results:** Figure 11, 12, 13, 14, 15, 16 show additional de-noising results for image data and human skeleton pose data.

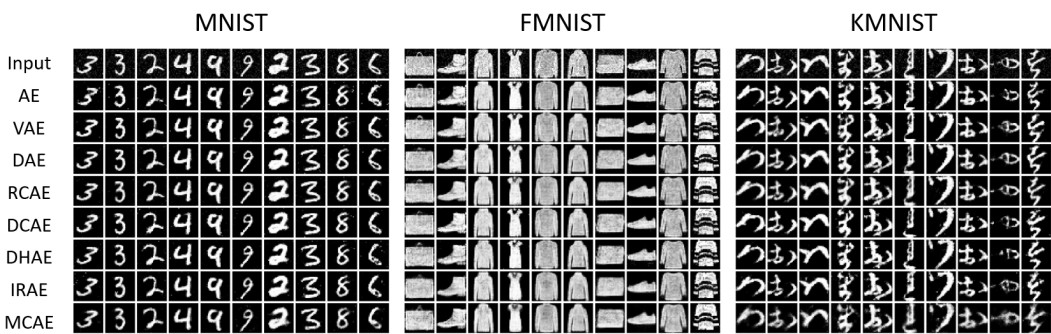

Figure 11: De-noising examples of grayscale image data (noise level 0.1).

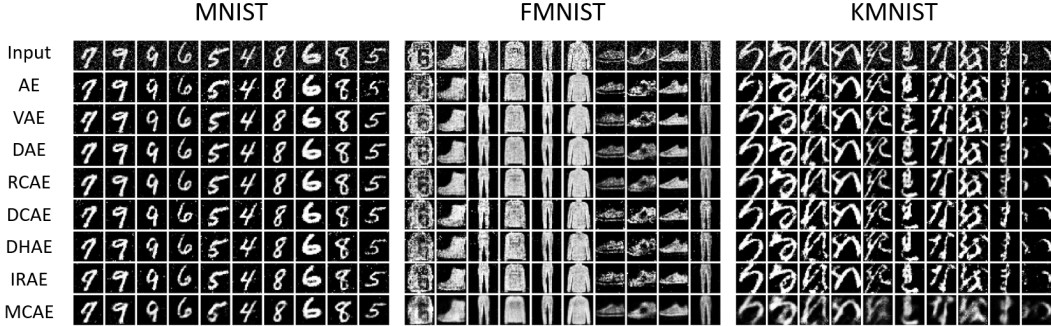

Figure 12: De-noising examples of grayscale image data (noise level 0.2).

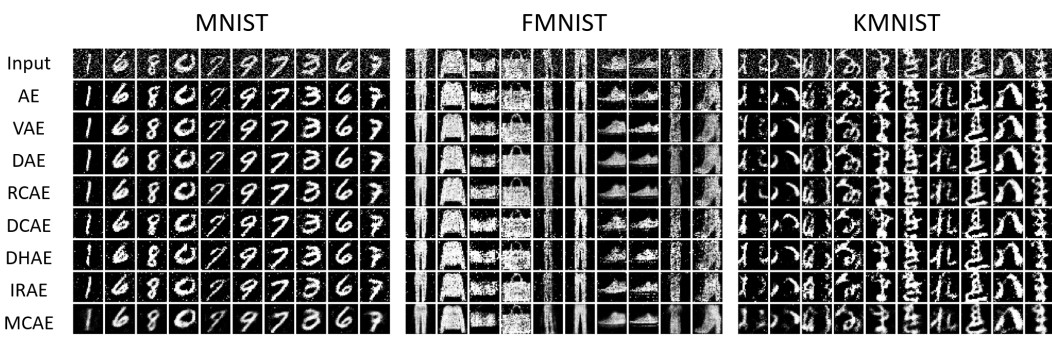

Figure 13: De-noising examples of grayscale image data (noise level 0.3).

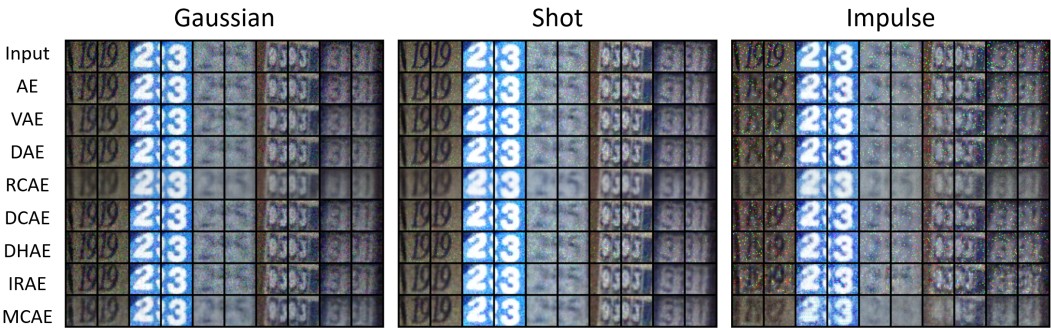

Figure 14: De-noising examples of SVHN data.

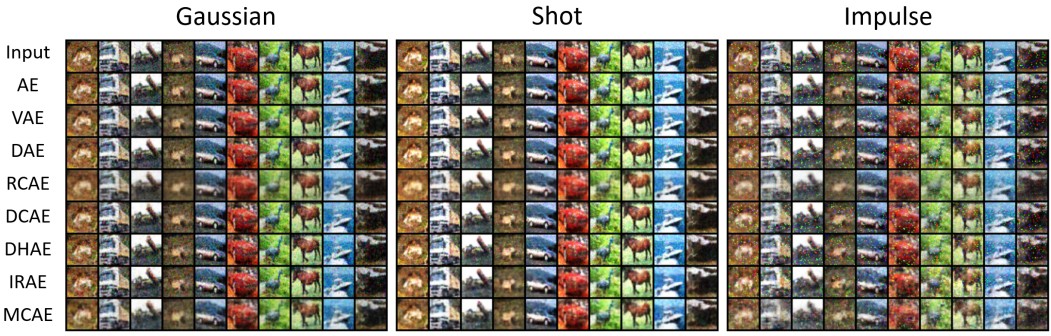

Figure 15: De-noising examples of CIFAR10 data.

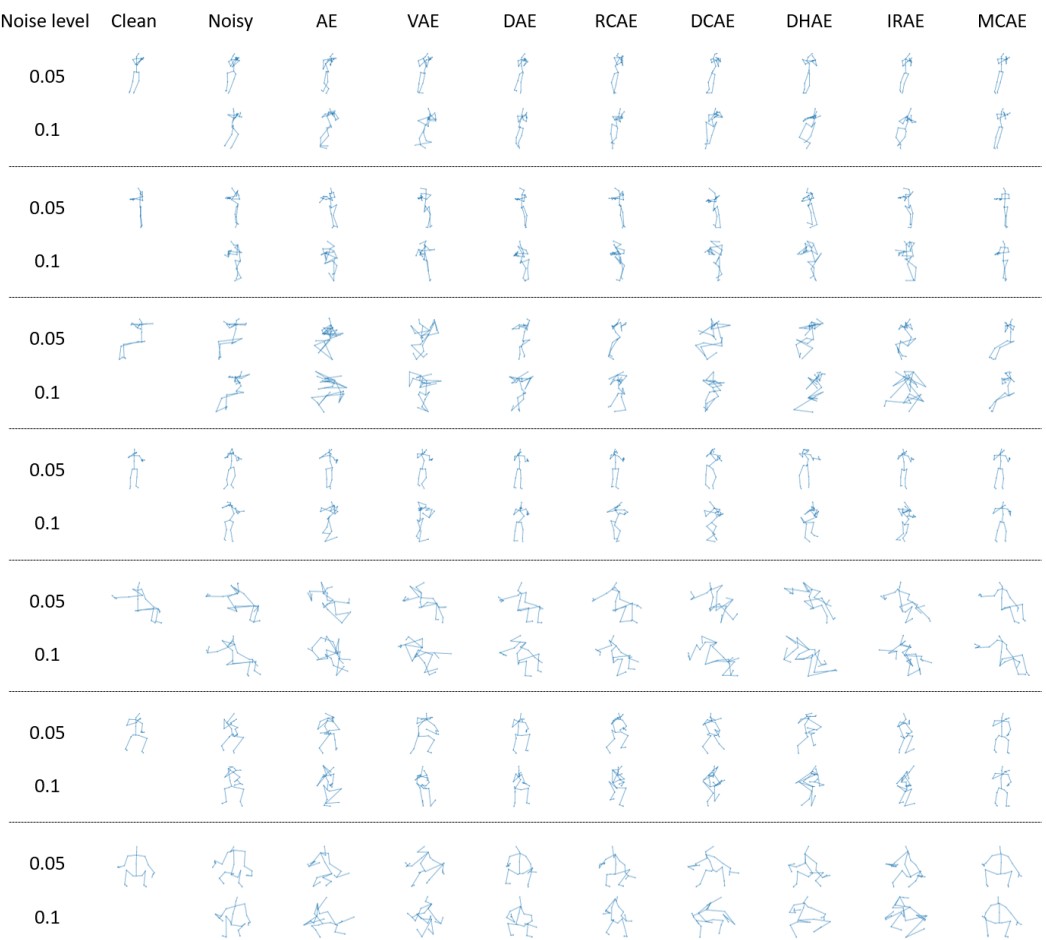

Figure 16: De-noising examples of human pose data.

## A.6 Computational Complexity

In this section, we provide actual computation time of the curvature measure in (7) and backpropagation time with two-layer fully connected neural networks used for gray-scaled image data and convolutional and transposed convolutional neural networks used for the SVHN and CIFAR10 image data. Throughout this study, the NVIDIA GeForce RTX 3090 is used.

Table 5: Per-batch computation time comparisons. For the FC net with $1 \times 28 \times 28$ image, the latent space dimension is 16 and the batch size is 100, and for the Conv net with $3 \times 32 \times 32$ image, the latent space dimension is 64 and the batch size is 8 (for GPU memory limitation).

| | FC net with $1 \times 28 \times 28$ image | | Conv net with $3 \times 32 \times 32$ image | |
| --- | --- | --- | --- | --- |
| | Forward Computation | Back-Propagation | Forward Computation | Back-Propagation |
| Reconstruction | 0.00037 s | 0.00045 s | 0.00220 s | 0.00114 s |
| Curvature | 0.00967 s | 0.00447 s | 0.30147 s | 0.06051 s |

Table 6: Per-batch computation times of the intermediate operations in curvature measure (7) with the Conv net with $3 \times 32 \times 32$ image and 64-dimensional latent space.

| $J_{f_\theta}^T J_{f_\theta}$ | $G_\theta^{-1}$ | $\frac{\partial(E(J_\theta)w)}{\partial z}v$ | $\frac{\partial(E(J_\theta)w)}{\partial z}(G_\theta^{-1}v)$ |
| --- | --- | --- | --- |
| 0.00192 s | 0.28353 s | 0.00682 s | 0.00647 s |

Table 7: Per-batch computation times and percent errors of the approximate matrix inverse $G_\theta^{-1}$ in curvature measure (7) as the number of iteration increases with the Conv net with $3 \times 32 \times 32$ image and 64-dimensional latent space.

| number of iterations | ground truth | 1 | 5 | 10 | 100 | 1000 |
| --- | --- | --- | --- | --- | --- | --- |
| time | 0.28353 s | 0.000256 s | 0.000507 s | 0.000807 s | 0.005669 s | 0.053519 s |
| percent error | 0 % | 99.83 % | 97.69 % | 71.35 % | 0.0109 % | 0.0095 % |

Table 5 shows the per-batch computation time comparisons between reconstruction loss term and curvature term in (7). Although the curvature computation has become feasible through the stochastic trace estimation, compared to the original reconstruction loss term, it still takes much longer time. Especially, looking at the forward computation time for the Conv net case, the curvature computation is almost 100 to 150 times slower than the reconstruction term computation.

To see which part in the below curvature measure

$$\mathcal{C}(\theta,\phi) = \mathbb{E}_{z\sim\hat{p}_\phi(z),v\sim\mathcal{N}(0,I_m),w\sim\mathcal{N}(0,I_D)}\left[v^T\frac{\partial(w^T E(J_\theta))}{\partial z}\frac{\partial(E(J_\theta)w)}{\partial z}G_\theta^{-1}v\right]$$

requires a major computational cost, we compare the computation times of the following operations: (i) the Riemannian metric $G_\theta = J_{f_\theta}^T J_{f_\theta}$, (ii) the inverse of $G_\theta$, (iii) the Jacobian-vector product for $\frac{\partial(E(J_\theta)w)}{\partial z}v$, and (iv) the Jacobian-vector product for $\frac{\partial(E(J_\theta)w)}{\partial z}(G_\theta^{-1}v)$.

Table 6 shows the per-batch computation times of the intermediate operations in curvature measure (7) for the Conv net case. As can be seen, the inverse computation takes up most of the total computation time. To reduce the computation time of the matrix inverse, one can consider an approximate inverse computation method. For example, given $G_\theta$, let us define a function $f : \mathbb{R}^{m\times m} \to \mathbb{R}^{m\times m}$ such that

$$f(X) = X^{-1} - G_\theta. \tag{15}$$

To find the root of $f$, we can use the standard Newton-Raphson method:

$$X_{n+1} = 2X_n - X_n G_\theta X_n, \tag{16}$$

which is known as the Newton-Schulz iteration method for the matrix inversion. We can get an approximation of $G_\theta^{-1}$ by iteratively applying the above, where it gets closer to the true inverse as we increase the number of iteration.

Table 7 shows the per-batch computation times and percent errors of the approximate matrix inverse $G_\theta^{-1}$ in curvature measure (7) as the number of iteration increases with the Conv net case. The percent error is computed as $100 * \|G_{\text{true}}^{-1} - G_{\text{est}}^{-1}\|_F/\|G_{\text{true}}^{-1}\|_F$. When the number of iterations is set to be 100, the percent error is only 0.01 % while significantly reducing the computation time as 0.28353 s $\to$ 0.005669 s. It is highly recommended to use the approximate matrix inverse when the latent space dimension is high.

