# OpenReview forum: "Minimum Curvature Manifold Learning"
_ICLR.cc/2023/Conference — Submitted to ICLR 2023_

### Official Review · Reviewer_idbz · 2022-10-20

**Confidence:** 3
**Correctness:** 4
**Technical Novelty And Significance:** 2
**Empirical Novelty And Significance:** 2
**Recommendation:** 5

**Clarity, Quality, Novelty And Reproducibility:**

Basically the paper is readable, however the readibility should be increased.  For example, how equation (5) implements the Dirichlet energy in (4) when consider the mappy T(z).

For reproducibility:   The author(s) might submit their experiment codes

**Strength And Weaknesses:**

Strength:

1. Take the view of focusing on regularizing decoder in an autoencoder framework. This is implemented through the newly introduced minimum extrinsic curvature principle for manifold regularization, which is based on the Dirichlet energy for mappings between Riemannian manifolds, while they specially look at the latent space and the Grassmann manifold of the data manifolds in the data space.

2. The new manifold regularizer is introduced into the classic autoencoder objective to form the minimum curvature autoencoders

3. Practical implementation issue has been carefully attained.

Weakness:

1. As the paper mentions, the intrinsic curvature does not capture how the manifold lies in the data space, particularly in dimension higher than 2. Although the new definition of the curvature generalizes classical definition of the curvature of a curve embedded in R^3, it is not clear whether the new measure has better regularization power given that less geometry information is used, for example, torison and/or tensor on manifold.   I know this is not actionable point, but wish to point out.

2. Perhaps an experiments in highdimensional data set.

3. Better readibilty, some details should be added in, see below.

**Summary Of The Paper:**

Keeping the coordinate-invariant property in mind, the author(s) propose the minimum extrinsic curvature principle for manifold regularization and a Minimum Curvature Autoencoder.   The main focus is to take an appropriate regularization for the decoder function in the autoencoder, as mentioned in the paper as the decoder has information about how the manifold lies in the data space. They introduce a coordinae-invariant extrinsic curvature measure by investigating how smoothly tangent space changes on the manifold and use it as a regularizer.  Under this new regularized, the experiments have shown improved manifold learning accuracy for both noising and small training datasets.

**Summary Of The Review:**

Overall the paper has its merit in novelty. It is basically and logically clear with convincing experiments.

---

### Official Review · Reviewer_xNDN · 2022-10-24

**Confidence:** 4
**Correctness:** 3
**Technical Novelty And Significance:** 3
**Empirical Novelty And Significance:** 2
**Recommendation:** 6

**Clarity, Quality, Novelty And Reproducibility:**

The paper is clear and easy to read. The only exception is that I think the used approximation of the metric inverse should be part of the main paper. The method is sensible and novel (as far as I can tell). Regarding reproducibility, then it would be good if the authors released their code (I would look positively upon a statement of this in the rebuttal). I think the future impact of this work depends on how well it will scale to large models, which is currently not explored in the paper.

**Strength And Weaknesses:**

## Strengths
* I found the paper to be well-motivated and well-written.
* The proposed measure of curvature seems both novel and well-founded.
* I found the computational approximation used to numerically evaluate the curvature measure to be interesting and novel. (see also Weakness below)

## Weaknesses
* The main paper presents the "theoretical" way to evaluate the curvature measure, while the "practical" implementation is only described in the appendix. I understand that space is limited, but it feels like this aspect of the work is being hidden (does it not work well?). The idea of replacing the inverse metric with a Gauss-Newton approximation is, not entirely novel, but within the context, I find it novel. As such, I really think this part of the work should be in the main paper, and further investigated. I really miss an empirical evaluation of the influence of this approximation. Does it not depend on an initial guess of the inverse metric (i.e. a preconditioner)? The appendix indicates that a single step of Gauss-Newton is sufficient, but is it really? I'd like to see an empirical study here. In general, I think this part of the work is too important to brush aside and not give a more thorough study.
* I am somewhat skeptical about the idea of using mix-up as a way to evaluate the expectation over the latent representations. I understand the motivation and trust that this is probably a sensible idea. However, this mix-up is also a regularize on its own, which then makes it difficult for me to determine the importance of the curvature regularize as you now have two regularization schemes working in parallel. I really miss some sort of ablation study here showing that the regularization induced by mix-up is not what is driving the empirical results. Without this, I find it difficult to judge the presented empirical results.
* In general, the empirical results are limited to quite small models. This is understandable for a paper of this type, but it is still a limiting factor. It would be good to demonstrate that the approach can scale to large models.

## Minor things:
* In the introduction, the term "graph" is used. I suppose this refers to the image of the latent space under the decoder, but it would be good to be explicit about what is exactly meant. I can easily imagine confusion caused by some readers thinking about discrete grasp (collections of nodes and edges).
* In the weaknesses part of the conclusion it is mentioned that it would be good to adapt the regularization strength locally. A recent paper does exactly this, but uses a notion of geometric reach rather than curvature (those are tightly linked topics). https://arxiv.org/abs/2206.01552 It would be good to include this line of work in the related work section.
* In the last sentence of the first paragraph of section 2.2, you should write $f$ rather than f.
* The reference to "LEE et al." should be "Lee et al."
* While I appreciate the opening example of how Jacobian regularizers are sensitive to volume changes, I think this particular example is not entirely convincing as many people, in practice, place Gaussian priors over the latent representations thereby "fixing" the volume issue (at least in practice). I get the theoretical argument, but I wonder if another example could be used?
* In definition 1, does h have to be a smooth function as well (diffeomorphism)?
* The last sentence of page make it sound like alpha is called the MCAE. I propose to rephrase.



**Summary Of The Paper:**

The paper proposes an extrinsic measure of the curvature of the manifold spanned by a decoder. This measure is then used in an autoencoder as a regularizer. In practice, the curvature measure is approximated to limit computational costs. Empirical results are limited to small models, but results appear promising.

**Summary Of The Review:**

The paper has a clearly explained and novel contribution, with a somewhat unclear empirical impact. I vote in favor of acceptance, but the empirical aspect leave me only luke-warm.

---

### Official Review · Reviewer_fxZi · 2022-10-25

**Confidence:** 3
**Correctness:** 3
**Technical Novelty And Significance:** 2
**Empirical Novelty And Significance:** 1
**Recommendation:** 3

**Clarity, Quality, Novelty And Reproducibility:**

Good clarity, acceptable novelty. Quality of results is not the best and I suspect good reproducibility

**Strength And Weaknesses:**

Strengths

- The paper reads well and the arguments in the motivation seem persuasive
- I found the curvature construction interesting, although I am not fully sure how novel it is (see weaknesses)
- Figure 3 to me was the highlight of this paper, where the proposed method achieves lower reconstruction errors for the same degree of curvature reduction

Weaknesses

- Unfortunately, the practical benefits of this paper seem very limited - even in the context of auto-encoder manifold learning methods. Tables 2, 3 and 4 do not post a convincing picture and when I put that together with Figure 3(a) I find it hard to argue for a substantial practical gain of this method with IRAE since it appears even IRAE minimizes the curvature without really modeling it in the proposed way.
- I think the authors must provide more historical/literature background for the curvature definition for manifolds. For eg, how is the proposed formulation related to Rici curvature? and why minimizing that is not beneficial? Despite the minor references, a direct formulation of curvature without any such background feels odd.
- Figure 4 is not clear. Please provide a more clear visualization of where the manifold are badly approximated

Questions and Suggestions

- Could there be an experimental demonstration of the need for reparametrization invariance? Conceptually it makes sense, but I don’t fully see why it should be a big deal in practice (I suppose overfitting to the discretization of the manifold? but how bad is it really?) The same applies to “extrinsic” measures. It feels a bit too less to only have a diagram in figure 1 and not something concrete in practice. I make this remark because demonstrating such a case could provide more meat to the contribution of this paper, especially in contrast to IRAE



**Summary Of The Paper:**

This paper investigates an unsupervised learning/manifold learning paradigm focusing on the auto-encoder framework and proposes a new regularization formula for the latent space. The authors develop a notion of curvature for the manifold that is extrinsic yet reparameterization invariant which they argue is desirable in contrast to some recent similar prior works in the area. Conceptually the curvature formulation measures the integral of an infinitesimal change in a tangent space at a point on the manifold and utilizes the Grassman manifold structure of linear subspaces.

Experimentally, the authors compare 3 different scenarios, 1 synthetic, image (MNIST, SVHN, CIFAR10) and motion capture dataasets and focus on comparing with only other auto-encoder methods. The results seem to suggest that the proposed curvature minimization in the auto-encoder framework yields a degree of robustness (though not explicitly trained).

**Summary Of The Review:**

All in all. I am inclined to weigh in negatively, mainly because I am not fully convinced of the efficacy of the results. Despite that, I did find the paper to be an interesting read and wish the authors can elaborate more on the seemingly novel and interesting aspects of their contribution namely - the desire for extrinsic curvature and reparameterization invariance.

---

### Decision · Program_Chairs · 2023-01-20

**Decision:**

Reject

**Justification For Why Not Higher Score:**

N/A

**Justification For Why Not Lower Score:**

N/A

**Metareview: Summary, Strengths And Weaknesses:**

Thank you for submitting your work to ICLR 2023.
The consensus among reviewers was that this work is not ready for publication. The reviewers did find the work interesting with some novel and potentially useful ideas. The main issues are that the empirical justification for this method was lacking and not sufficient to demonstrate the efficacy of the method, and relating the proposed approach to existing regularization/curvature measures.